# Electrostatic forces alter particle size distributions in atmospheric dust

Joseph R. Toth III[1*], Siddharth Rajupet[1*], Henry Squire[1], Blaire Volbers[1], Jùn Zhou[2,3], Li Xie[2,3], R. Mohan Sankaran[1], Daniel J. Lacks[1]

[1]Department of Chemical and Biomolecular Engineering, Case Western Reserve University, Cleveland, OH 44106
[2]Key Laboratory of Mechanics on Disaster and Environment in Western China, Ministry of Education of China, Lanzhou, Gansu, 7300000, China
[3]College of Civil Engineering and Mechanics, Lanzhou University, Lanzhou, Gansu, 7300000, China
[*]Authors contributed equally to paper

*Correspondence to*: Daniel J. Lacks (djl15@case.edu)

**Abstract.** Large amounts of dust are lofted into the atmosphere from arid regions of the world before being transported up to thousands of kilometers. This atmospheric dust interacts with solar radiation causing changes in the climate, with larger-sized particles having a heating effect, and smaller-sized particles having a cooling effect. Previous studies on the long-range transport of dust have found larger particles than expected, without a model to explain their transport. Here, we investigate the effect of electric fields on lofted airborne dust by blowing sand through a vertically-oriented electric field, and characterizing the size distribution as a function of height. We also model this system, considering the gravitational, drag, and electrostatic forces on particles, to understand the effects of the electric field. Our results indicate that electric fields keep particles suspended at higher elevations and enrich the concentration of larger particles at higher elevations. We extend our model from the small-scale system to long-range atmospheric dust transport to develop insights on the effects of electric fields on size distributions of lofted dust in the atmosphere. We show that the presence of electric fields and the resulting electrostatic force on charged particles can help explain the transport of unexpectedly large particles and cause the size distribution to become more uniform as a function of elevation. Thus, our experimental and modelling results indicate that electrostatic forces may in some cases be relevant regarding the effect of atmospheric dust on the climate

## 1 Introduction

The global climate is largely governed by a radiative balance between incoming solar and outgoing thermal radiation from the Earth (Hansen, 2005). This balance depends on interactions between matter and radiation. Dust represents an increasingly significant amount of matter in the atmosphere and interacts with radiation both directly, through scattering and absorption (Tegen, 2003; Kok et al., 2018) and indirectly, by impacting processes such as cloud formation and dissipation and biogeochemical feedbacks (Mahowald, 2011). The source of this dust is typically arid regions such as northern Africa and central Asia, where wind ejects dust into the atmosphere (Bagnold, 1971; Kok et al., 2012). Atmospheric dust can travel far from its source (Betzer et al., 1988) – e.g., Saharan dust travels to the western hemisphere – and so dust events in one region can affect climate in distant regions. Anthropogenic factors such as over-cultivation and water diversion significantly increase the amount of atmospheric dust (Ginoux et al.,

2012; Webb and Pierre, 2018). Thus, accurately characterizing the effects of atmospheric dust on the Earth's radiative balance is necessary for understanding the anthropogenic role in climate change.

The size distribution of dust systems has been shown to have important implications on the climate (Mahowald et al., 2014). Scattering and absorption of different wavelengths of light depend on the size distribution of airborne particles (Tegen and Lacis, 1996). Smaller-sized particles predominately scatter solar radiation, which has shorter wavelengths, away from the Earth, causing a cooling effect on the climate (Kok et al., 2017). In contrast, larger-sized particles predominately scatter thermal radiation emitted by the Earth, which has longer wavelengths, back towards the earth, causing a heating effect on the climate. In addition, larger particles are more effective at absorbing both solar radiation and thermal radiation from the earth, trapping heat in the atmosphere (Otto et al., 2007). The net effect of atmospheric dust on the climate thus depends on the particle size distribution of the dust.

Previous field studies have found surprising results regarding the particle size distributions of dust transported far from the source (Betzer et al., 1998; Maring et al., 2003; Reid et al., 2003; Haarig et al., 2017; Renard et al., 2018; van der Does et al., 2018). The distributions contain larger particles than predicted by models, which, as described above, could have implications on the climate. Various mechanisms could be responsible for this effect including electrostatic forces (Ulanowski et al., 2007; van der Does et al., 2018).

It has been known for over 100 years that dust particles can become highly charged, as indirectly observed by the presence of strong electric fields in dust storms (Rudge, 1913). These electric fields can be large, with magnitudes greater than 100 kV m$^{-1}$ (Schmidt et al., 1998). The electric fields develop as follows. Wind-blown dust particles collide with one another and transfer charge by triboelectric charging; this process is the same as that occurring when a balloon is rubbed on hair. Triboelectric charging occurs even when all particles are composed of the same material, with some particles charging positive and others charging negative (Lacks and Sankaran, 2011; Xie et al., 2013). Due to the cancelling contributions of positive and negative particles, the volumetric charge density in lofted dust systems is expected to be very small, while the charge on each individual particle can still be large. Studies have shown that, to some extent, smaller and larger particles tend to charge with different polarities (Zhao et al., 2002; Forward et al., 2009; Bilici et al., 2014; Waitukaitis et al., 2014; Toth et al., 2017). This particle-size-dependent polarity of charge, combined with the separation of small and large particles in a gravitation field such that smaller particles are lofted higher, generates the electric fields in dust storms.

The electric fields generated in dust storms have been shown to cause dust liftoff, i.e., transferring particles from the ground to the air (Kok and Renno, 2006, 2008). The dust liftoff was subsequently confirmed by wind tunnel experiments (Rasmussen et al., 2009). Recently, field studies of dust storms found a strong correlation between the electric field and the density of airborne particles (Esposito et al., 2016), which supports the prediction that electric fields lead to dust liftoff.

In this paper, we address the effects of electrostatics on lofted dust, i.e. particles that are already airborne. The electric fields leading to these effects could be created in dust storms, as described above, or could occur in the atmosphere for other reasons (e.g., fair weather electric fields). The effect of electric fields on airborne dust are fundamentally different than those on dust liftoff, where physical contact with the ground surface allows charge transfer from the particles to (electrical) ground. To reveal how electric fields affect airborne dust, we carried out

controlled laboratory experiments in which electric fields are applied to lofted particles and characterized the particle size distributions as a function of the lofted height. We also develop a model to support our experimental results, and then apply the model to prior studies to elucidate the role of electric field on long-range dust transport.

## 2 Experimental Methods

We constructed a setup to characterize the effects of an electric field on dust trajectories after lift-off. The setup, shown in Fig. 1, consisted of two parallel electrodes (89 cm long, 15 cm wide) connected to a DC high-voltage power supply (HB-Z303-1AC) oriented such that the electric field between the electrodes was perpendicular to the ground surface. The distance between the electrodes was 12 cm, and the electric field was varied from 125 to -125 kV m$^{-1}$ where the polarity of the electric field was defined by the polarity of the top electrode; i.e., a positive electric field had a positive top electrode and a negative electric field had a negative top electrode. In order to prevent particle contacts with the electrodes and ensure that the sand particles contacted only other sand particles (i.e., not other surfaces), the electrodes were covered with a thin layer of sand held in place by two-sided tape.

A 250 g sand bed 1.5 cm tall, 20 cm long, and 11 cm wide was positioned 4 cm upstream of the electrodes. The sand was polydisperse and characterized by a mean diameter of ~132 μm, 40% by mass with diameters smaller than 105 μm, and 60% by mass with diameters between 105 and 450 μm. Particles with diameters greater than 450 μm were removed by sieving. A fan placed 15 cm upstream of the sand bed was used to blow sand through the electrodes. The average air speed over the sand bed was 6.7 m s$^{-1}$, and decreased to 3.2 m s$^{-1}$ at the end of the electrodes. As it is difficult to control exactly when particles become charged, the particles likely had an initial charge in the sand bed due to particle contacts during sample preparation and could acquire more charge during lift-off and transport between the electrodes. Each trial ran for 2 minutes, consuming approximately 80% by mass of the sand bed.

To collect particles, two cups were placed 11 cm downstream the electrodes with slit openings 2.2 cm tall and 11 cm wide. The bottom cup collected particles between heights of 0 and 2.2 cm, and the top cup collected particles between heights of 8.8 and 11 cm. The total mass collected in both cups in the experiments ranged from 8 g to 35 g, which represents approximately 21% of the mass passed through the system by the fan. The particles collected in the two cups were sieved with a 105 μm mesh sieve; "small particles" were defined as those that passed through the sieve and "large particles" were defined as those that did not pass through. After sieving the particles in each cup, we determined the mass of the small and large particles collected in each cup, $m_i^j$, where $i$ designates the particle size ($i = S$ for small particles, $i = L$ for large particles) and $j$ designates the cup position ($j = T$ for top, $j = B$ for bottom).

## 3 Modeling Methods

We developed a Monte Carlo model to address the effects of electric fields on dust transport in both our experimental system as well as previous field studies of dust transported long distances in the atmosphere. Our simulations address the final positions of wind-blown particles under various conditions. We assumed the particles are spheres of diameter, $D$, with density, $\rho$, and surface charge density, $\sigma$. In the horizontal direction, we do not consider forces on the particles

and we assumed that the particles travel at the velocity of the wind. In the vertical direction, we assumed that the only

forces acting on the particles are gravitational, electrostatic, and drag forces. The gravitational force, $F_G$, is given by,

$$F_G = \frac{\pi D^3}{6} \rho g \tag{1}$$

where $g$ is the acceleration due to gravity. The electrostatic force, $F_E$, is given by,

$$F_E = \pi D^2 \sigma E \tag{2}$$

where $E$ is the electric field strength. The drag force, which always acts in the direction opposite to vertical velocity,

is given by,

$$F_D = -\frac{1}{8} C_D \rho_f \pi D^2 v^2 \hat{v} \tag{3}$$

where $\rho_f$ is the density of air, $v$ is the velocity of the particle in the vertical direction, and $\hat{v}$ is the direction of motion

of the particle. We neglected air velocity in the vertical direction. For the drag coefficient, $C_D$, we used the following

correlation (Brown and Lawler, 2003),

$$C_D = \frac{24}{Re}(1 + 0.15 Re^{0.681}) + \frac{0.407}{1 + \frac{8710}{Re}} \tag{4}$$

where $Re$ is the Reynolds number,

$$Re = \frac{\rho_f |v| D}{\eta} \tag{5}$$

and $\eta$ is the viscosity of air. We note that the drag and electrostatic forces can act upwards or downwards depending

on the direction of particle velocity and the sign of $\sigma E$ respectively. The net force on a particle, $F_{net}$, is the sum of $F_G$,

$F_E$, and $F_D$.

    The equations governing the motion of the particle in the vertical direction are

$$\frac{dv}{dt} = \left(\frac{6}{\pi D^3 \rho}\right) F_{net} \tag{6}$$

and

$$\frac{dy}{dt} = v \tag{7}$$

where $y$ is the position of the particle in the vertical direction.

    Particle trajectories are determined by numerically integrating Eqns. (6) and (7). At sufficiently long times,

the system attains a constant terminal velocity – i.e., as the particle velocity increases due to acceleration from

gravitational and electrostatic forces, the drag force increases until it balances other forces and $F_{net} = 0$. Thus, steady-

state particle trajectories can be obtained by calculating the terminal velocity, rather than by integrating the equations

of motion.

    Since particles traverse the electrodes more quickly (18 ms) than terminal velocity can be reached in our

laboratory experiments, particle trajectories were obtained by numerical integration of the equations of motion. In the

horizontal direction, particles move at the average wind speed in the system (4.95 m s$^{-1}$). In the vertical direction,

particles start with size-dependent initial vertical velocities, at an initial height of zero. We considered a collection of

particles of density, $\rho = 2600$ kg m$^{-3}$ (characteristic of quartz), where the particle diameter, charge, and initial vertical

velocity, $v_0$, were chosen randomly from normal distributions characterized by respective means, $\mu$, and standard

deviations, $s$. For particle diameter, $\mu_D = 132$ μm and $s_D = 60$ μm (as in the experiments). For surface charge density,

$\mu_\sigma = 0$ and $s_\sigma = \alpha$, where $\alpha$ is a fitting parameter. For initial vertical velocity, $\mu_{v_0} = 0$ and $s_{v_0} = \beta/D^2$ that depends

on particle diameter, where $\beta$ is a fitting parameter (physically, this relationship causes smaller particles to have higher initial velocities); only positive initial velocities were sampled from the distribution. Particle trajectories were calculated using Euler's method with a time step of 18 µs. After traveling the length of the electrode (0.89 m), particles with heights between 0 and 6 cm were considered to be collected in the bottom cup, and those with heights greater than 6 cm were considered to be collected in the top cup. Particles with heights less than 0 cm were not considered to be collected in the collection cups.

For long-range dust transport, particles reach terminal velocity, and thus a simpler method can be used to determine particle trajectories. The dust was assumed to be already lofted and following the model of Maring et al. (2003), particles were uniformly distributed in a vertical 2000 m window, with initial velocities equal to their terminal velocities. We considered a collection of particles of density $\rho = 2600$ kg m$^{-3}$, and the particle diameter and charge were chosen based on the study being modeled. The terminal velocity was found by numerically solving for the velocity at which $F_{net} = 0$. The final height of each particle after traveling for a given amount of time was obtained simply by calculating the initial height plus the terminal velocity multiplied by time.

## 4 Laboratory Studies

First, we address the effect of the applied electric field on the elevation of airborne particles. We quantify this effect by the fraction of total collected particles in the top cup, $F^T$,

$$F^T = \frac{m_L^T + m_S^T}{m_L^T + m_S^T + m_L^B + m_S^B}. \tag{9}$$

Figure 2 shows our results for $F^T$ as a function of applied electric field. As the magnitude of the electric field increases, more of the airborne dust remains suspended higher, as indicated by the increase in the fraction of particles collected in the top cup. We note that this enhancement of dust elevation occurs with either polarity of electric field.

We now turn to the particle size dependence of the electric field effect. Figure 3(a) shows results for the fraction of large particles that are collected in the top cup rather than the bottom cup, $g_L^T$,

$$g_L^T = \frac{m_L^T}{m_L^T + m_L^B} \tag{10}$$

Likewise, Fig. 3(b) shows results for the fraction of small particles that are collected in the top cup rather than the bottom cup, $g_S^T$,

$$g_S^T = \frac{m_S^T}{m_S^T + m_S^B}. \tag{11}$$

The values $g_L^T$ and $g_S^T$ show the fraction of large and small particles, respectively, in the system that are collected in the top cup. When no electric field is applied, $g_S^T$ is greater than $g_L^T$ meaning that more of the small particles are in the top cup than of the large particles in the top cup. This is due to gravitational effects which cause smaller particles to remain suspended higher than larger particles. As the magnitude of the electric field increases, large and small particles both remain suspended higher, as shown by the increase in the fractions of large and small particles in the top cup.

We now examine how the size distribution of particles at higher elevations is affected by the electric field. The fraction of particles in the top cup that are large, $f_L^T$, is defined as

$$f_L^T = \frac{m_L^T}{m_L^T + m_S^T}.$$ (12)

While $f_L^T$ can be obtained directly from the experimental data, these results are especially susceptible to noise, as the overall ratio of large to small particles collected varies between trials. This could be due to the fact that the initial particle size distribution could change over time as small particles are preferentially lost from the system. We can get a better estimate of $f_L^T$ by calculating it independently, using variables that are less susceptible to noise,

$$f_L^T = \frac{g_L^T F_L}{F^T},$$ (13)

where $F_L$, the fraction of particles in both cups that are large, is defined as

$$F_L = \frac{m_L^T + m_L^B}{m_L^T + m_s^T + m_L^B + m_s^B},$$ (14)

$g_L^T$ and $F^T$ come from fits to our experimental data, and the average value of $F_L$ across all trials is $0.63 \pm 0.01$. The results for $f_L^T$ are shown in Fig. 4. We see that an electric field causes the particle size distribution at higher elevations to be enriched in larger particles.

We carried out simulations of our model for the experimental system. The fitting parameters were varied until the model showed a reasonable fit to the experimental results for $\alpha = 1.4\ \mu C\ m^{-2}$ and $\beta = 11000\ \mu m^2$. We note that surface charge densities characterized by $\alpha = 1.4\ \mu C\ m^{-2}$ are reasonable, as they are similar to results found experimentally in glass particle systems (Waitukaitis et al., 2014), and they are significantly smaller than surface charge densities measured for triboelectrically charged quartz (Miura and Arakawa, 2007; Shen et al., 2016). Results were obtained for trajectories of $10^6$ particles. As seen in Figs. 2-4, our model results match our experimental results reasonably well. In both our model and experiments, the electric field maintains particles at higher elevations and shifts the size distribution at higher elevations towards large particles. While the distributions for particle size and charge density are not perfectly representative of the particles in our experimental system, the trends in the model will be the same regardless of what distributions are used.

**5 Modeling Field Studies**

Since the effect of dust on the climate is dependent on particle size, to accurately gauge the role of atmospheric dust on the climate, the size distribution of airborne dust must be known. Our results indicate that electric fields can alter the size distribution of transported dust. As we describe below, previous field studies have found surprising results regarding particle size distributions of dust transported far from the source. Here, we applied our model to address whether these effects might be due to electrostatic forces on long-range dust transport.

Maring et al. (2003) compared the size distribution of aerosol samples collected in Izaña near the dust emission source with Puerto Rico after a transport time of 5.5 days from the source. In support, they modeled a 2000 m tall column of air typical of the Saharan Air Layer over the North Atlantic and Caribbean that has no vertical mixing and an initially uniform size distribution. When only taking into account gravitational and Stokes drag forces, their model predicted more large particles settled out than found in the transported aerosol samples. In contrast, their model showed agreement when they modified the terminal velocity with an arbitrary upward-contribution for all particles.

Using our model, we calculated the displacement of different-sized particles after transport for 5.5 days at terminal velocity to determine the fraction of particles removed due to settling using similar assumptions as Maring et al. (2003), but explicitly including electrostatic forces rather than an arbitrary upward velocity component. To simplify, we assumed that all the particles had the same polarity and magnitude of charge. As shown in Eq. 2, the electrostatic force on each particle depends on the particle surface charge density and ambient electric field strength. The actual surface charge density of lofted dust particles is unknown, but typical triboelectric charging values range over several orders of magnitude, from ~1 µC m$^{-2}$ (Lee et al., 2018; Waitukaitis et al., 2015; Wang et al., 2019), to ~10 µC m$^{-2}$ (Shen et al., 2016), and up to ~100 µC m$^{-2}$ (Cottrell, 1978; Donald, 1968; Horn and Smith, 1992; Matsuyama et al., 2003; Miura and Arakawa, 2007; Nordhage and Bäckström, 1977). The maximum electrostatic charge on the surface of a material prior to gas breakdown was found by Matsuyama (2018) to be approximately 400 µC m$^{-2}$ for a 10 µm particle. The magnitude of the electric field is also difficult to determine and can range over multiple orders of magnitude, with values up to 200 V m$^{-1}$ in fair weather electric fields (Adlerman and Williams, 1996; Bennett and Harrison, 2007; Harrison, 2011), up to 15 to 150 kV m$^{-1}$ in dust storms (Bo and Zheng, 2013; Harrison et al., 2016; Jackson and Farrell, 2006; Schmidt et al., 1998; Zhang et al., 2018), and up to 500 kV m$^{-1}$ during thunderstorms (Stolzenburg et al., 2007). Since both the charge density and the electric field can vary by orders of magnitude, we consider the product between the two, $\sigma E$, to be the important parameter for determining the effects of the electrostatic forces. Given the range of possible electric fields and charge densities discussed above, the value of $\sigma E$ can range from 0.2 mC V m$^{-3}$ to 200 C V m$^{-3}$; however, particles can only sustain a $\sigma E$ on the lower end of this range where there are more consistent electric fields and more prevalent charge densities. As shown in Fig. 5, when $\sigma E = 38$ mC V m$^{-3}$, our model results are found to match well with the experimental data of Maring et al. (2003).

Other field studies also found large particles to be enriched at higher elevations in comparison to models. Reid et al. (2003) used a particle measuring system mounted to an aircraft to measure dust size distributions in Puerto Rico and found that the ratio of large to small particles did not depend strongly on elevation. Furthermore, lidar studies of Saharan dust over Barbados (i.e., far from the source) suggested little variation in the particle size distribution between 1 and 4 km elevation, based on the near-constant value of the depolarization ratio and the similarity of the 355 nm, 534 nm, and 1085 nm laser results in this elevation range (Haarig et al., 2017).

We carried out simulations to investigate whether electric fields and the resulting electrostatic force on particles can lead to this enrichment of large particles at higher elevations. Particle diameters were based on a lognormal size distribution characteristic of atmospheric dust with $\mu_D = 1.5\,\mu$m and $s_D = 0.75\,\mu$m (Reid et al., 2003). The products of electric field and particle charge density were based on a normal distribution with $\mu_{\sigma E} = 0$ and $s_{\sigma E} = 0, 11,$ and 22 mC V m$^{-3}$ such that half the particles experienced an electrostatic force in the upward direction, and half in a downward direction. Simulations were carried out for $10^5$ particles. As shown in Fig. 6, in the case of no electrostatic forces, the average particle diameter is significantly larger at low elevations than at high elevations, in agreement with current models. In comparison, with $s_{\sigma E} = 22$ mC V m$^{-3}$, the average particle diameter becomes more constant with changing elevation.

Electrostatic forces act on charged particles such that those of one polarity are lifted to higher elevations while those of the opposite polarity fall to lower elevations. As a result, the elevation distribution of particles is

stretched out and becomes more uniform. This leveling of the elevation distribution occurs for all sized particles, such that both large and small particles are more uniformly vertically distributed, causing the size distribution of particles to become more constant with changing elevation.

Several field studies have found larger particles transported than predicted when only taking into account gravitational and drag forces (Betzer et al., 1988; van der Does et al., 2018; Renard et al., 2018). Some studies have found particles with diameters greater than 40 µm and ranging to 450 µm transported long distances (van der Does et al., 2018; Renard et al., 2018b). In our modeled system, if electrostatic forces are neglected, any particle with a diameter greater than 7.3 µm would settle out of the 2000 m column within 5.5 days. However, from Fig. 7, which represents the size distribution of lofted particles, we see that electrostatic forces cause several particles larger than 7.3 µm to remain lofted, up to 17.6 µm. Even with the consideration of electrostatic forces, it is not expected that these much larger particles (40 to 450 µm) would be lofted. Thus, it is likely that other forces contribute to the transport of large particles such as fast horizontal wind speeds, turbulence, and uplift in convective systems (van der Does et al., 2018).

For electrostatic forces to account for these deviations in size distributions of atmospheric dust particles, $\sigma E$ must be on the order of ~10 mC V m$^{-3}$. To the authors' knowledge, field studies of atmospheric dust have not measured surface charge densities of dust particles to confirm whether particles have sufficient charge for electrostatic effects to be significant; we note that volumetric charge densities have been measured (Nicoll et al., 2011), but cannot be translated to surface charge densities, which may be much larger than volumetric charge, which is the net charge of a distribution of negatively and positively charged particles. However, we emphasize that surface charge densities on the order of ~10 to ~100 µC m$^{-2}$, which could cause significant electrostatic effects on transport, are routinely observed in laboratory settings (Cottrell, 1978; Donald, 1968; Horn and Smith, 1992; Matsuyama et al., 2003; Miura and Arakawa, 2007; Nordhage and Bäckström, 1977; Shen et al., 2016). While charges on particles may decay through gas neutralization, experiments have found surfaces retaining charge even after days of exposure to ambient conditions (Olthuis and Bergveld, 1992; Yuan and Li, 2005; Leonov et al., 2006) and charged dust has been observed in the atmosphere even after being transported for long times (Nicoll et al., 2011; Harrison et al., 2018; Renard et al., 2018a). Moreover, subsequent particle collisions during transport could further charge particles through the triboelectric effect.

Additionally, while electric fields up to 200 V m$^{-1}$ are naturally occurring in the atmosphere (Adlerman and Williams, 1996; Bennett and Harrison, 2007; Harrison, 2011) and likely persist throughout transport, local electric fields within dust layers may be much higher due to the presence of dust particles and separation of positively and negatively charged particles (Brazenor and Harrison, 2005; Ulanowski et al., 2007). To the authors' knowledge, local electric fields within dust layers have not been measured over the course of long-distance dust transport. Thus, to better gauge the extent electrostatic forces influence dust transport, future field studies should investigate characteristic particle surface charge densities and local electric fields strengths within dust layers which can persist throughout long-distance dust transport.

**6 Conclusion**

We conducted experiments to characterize the size distribution as a function of height for sand blown between electrodes with vertically oriented electric fields and model this system considering the gravitational, electrostatic, and drag forces on particles. Our experimental and modeling results indicate that electrostatic forces maintain particles at higher elevations and increase the concentration of larger particles at higher elevations. We extend our model to long-distance dust transport and find that sufficient electrostatic forces suspend large particles that would otherwise settle out during transport, thereby increasing the concentration of large particles in atmospheric dust. Since large particles have a heating effect due to absorption and scattering of radiation, our results suggest that electrostatic forces could contribute to the warming of the climate. In addition, we show that sufficient electrostatic forces may explain unexpected size and elevation distributions of atmospheric dust. To better gauge the significance of electrostatic forces in atmospheric dust transport, future field studies should characterize particle surface charge densities and local electric fields subsisting in transported dust layers.

## Acknowledgements

This material is based upon work supported by the National Science Foundation under grant numbers 1559508, 1604909 and 1206480. We would like to acknowledge Andrew Wang, Amber Phillips, and Phwey Gil for their help in the preliminary design on the experiments.

## Author Contributions

D. J. L., R. M. S., L. X., J. Z., and J. R. T. helped to design the experiments that J. R. T., S. R., H. S., and B. V. carried out. D. J. L. and S. R. designed the model and simulations that S. R. carried out. D. J. L., S. R., and J.R. T. analyzed and discussed the results. S. R., D. J. L., and J. R. T. wrote the manuscript. J. R. T., S. R., H. S., B. V., and D. J. L. contributed to the editing of the paper.

## Competing Interests

The authors declare that they have no conflict of interest.

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

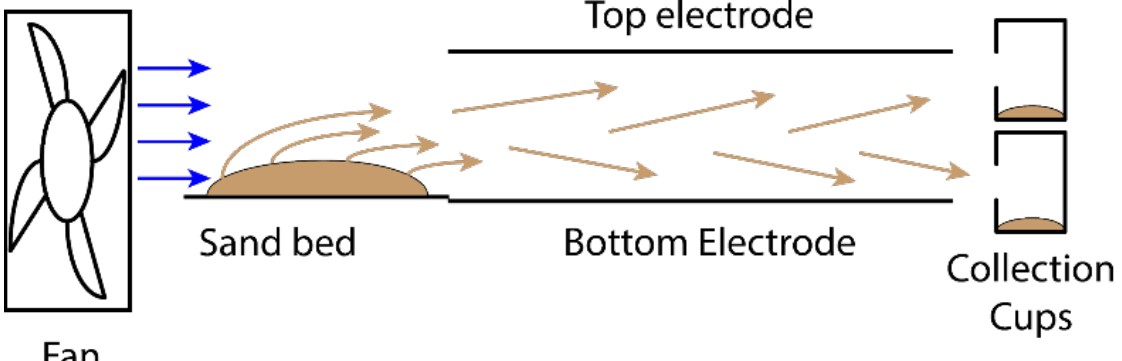

**Figure 1: An experimental schematic of the dust system. A fan blows sand from a bed between two electrodes. Sand is**
**collected in two collection cups after the electrodes.**

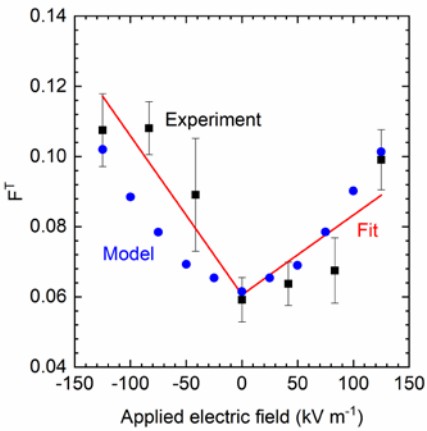

**Figure 2: Plot of the fraction of total particles collected in the top cup ($F^T$) as a function of applied electric field. The black squares are the experimental results, the red line is a linear fit to the experimental data, and the blue circles are data from the model.**

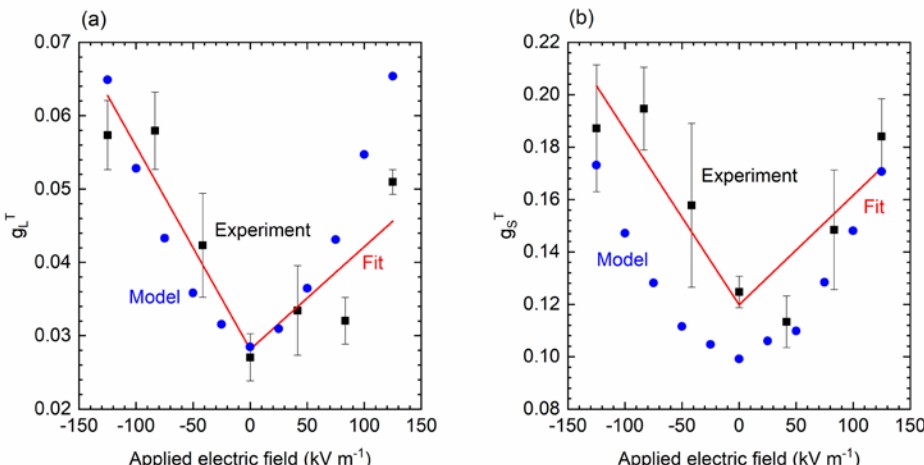

**Figure 3: Fraction of total (top and bottom) large particles in the top cup ($g_L^T$) (a) and fraction of total (top and bottom) small particles in the top cup ($g_S^T$) (b) as a function of applied electric field. The black squares are the experimental results, the red line is a linear fit to the experimental data, and the blue circles are data from the model.**

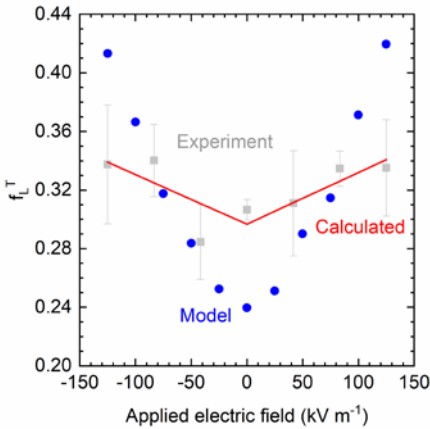

**Figure 4: Fraction of large particles in the top cup ($f_L^T$). The black squares are the experimental results, the red line is the calculated values of $f_L^T$ from Eq. (13), and the blue circles are data from the model.**

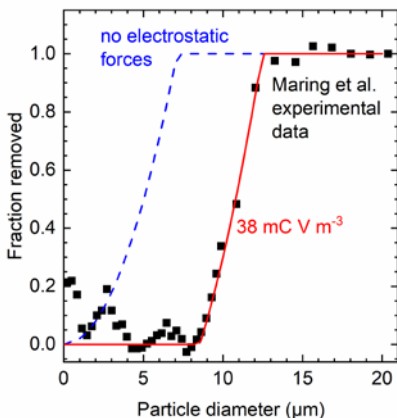

**Figure 5: Fraction of particles removed by settling. The dashed blue line takes into account gravitational and drag forces, the red line takes into account gravitational, drag, and upward electrostatic forces, and the black points are experimental data collected from Maring et al. (2003).**

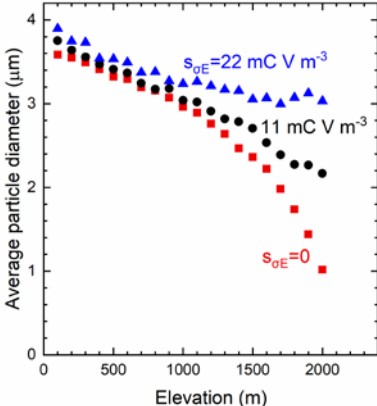

**Figure 6: Model results for the average particle diameter as a function of elevation after 5.5 days of transport at various standard deviations of electric field and particle charge density products, $s_{\sigma E}$.**

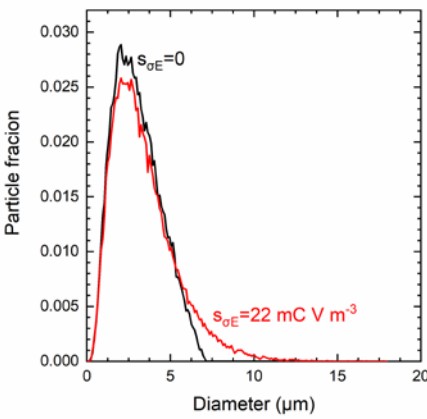

**Figure 7: Remaining particle size distributions within a 2000 m window after 5.5 days of transport at different values of $s_{\sigma E}$, with a bin size of 0.1 μm. The maximum size of lofted particles are 7.3 and 17.6 μm for a $s_{\sigma E}$ of 0 and 22 mC V m⁻¹, respectively.**

