# Peer review of "Electrostatic forces alter particle size distributions in atmospheric dust"

_Atmospheric Chemistry and Physics, 2019_

## Referee Comment (RC1) · Anonymous Referee #1 · 10 Sep 2019

The article presents a valuable contribution to the discussion of the influence of atmospheric electric fields on dust transport, on the basis of both lab measurements and modelling. Particularly interesting is the comparison of the modelling to the campaign data collected by Maring et al. (2003).

However, some improvement to the presentation is needed.

P.4 line 138 and following: the model description is cryptic and confusing, with the same symbols used for the mean values and standard deviations of three different quantities, diameter, charge density and initial vertical velocity. E.g. the variable described first as the standard deviation of the size distribution has the dimension of surface charge density (line 187). This makes it unnecessarily difficult to assess the methodology.

[Figure]

Another potential confusion occurs in section 5: on p.7 lines 211-212 it is stated that the electrostatic forces were calculated as the product of the electric field and the surface charge density, meant here to be the same charge density as that on the particle surfaces. Taken at face value, this might be understood to imply that the atmospheric system is treated as a parallel capacitor, with the charge density equal to that found on particle surfaces (or on bulk quartz surfaces, as in one cited reference, Miura and Arakawa, 2007). This is misleading at best, and could potentially hide a basic error made by the authors - which is presumably not the case, judging from equation 2, which is correct.

Having said that, there is some justification for employing the field-density product, as both these parameters are unknown and the electrostatic force does depend on their product. So it should be explained clearly what is meant.

My main objection concerns the discussion of the results. It is concluded that even the fair weather electric could maintain large dust particles aloft. However, this is assuming extreme value of surface charge densities, $220\,\mu C\,m^{-2}$, found in lab studies of bulk quartz surfaces (Miura and Arakawa, 2007), and untypically large (compare e.g. the values cited in Angus and Greber, 2018). More realistic values, especially for particulate matter rather than bulk material, should be cited from literature (which is not as sparse as the discussion implies, if spatial charge density measurements are included) and used in these conclusions. The authors own lab results point to charge densities two orders of magnitude lower than the above number, and similar values were found by Waitukaitis et al. (2014). So it should be concluded that electric fields exceeding the normal fair weather value are probably needed, as suggested previously by some authors. For example, it was estimated on the assumption of realistic spatial charge densities that fields of 2 kV/m would be required to maintain dust in suspension (Ulanowski et al., 2007). While such high values exceed the fair weather field, extensive literature suggests that they are not uncommon during dust episodes.

These unsupported conclusions are made prominent in Fig. 6 and 7, where the earlier

convention to use the field-density product (as in Fig. 5) is unaccountably abandoned in favour of citing just the electric field. The hidden assumption, not mentioned in the figure captions, is the extreme, and probably unrealistic, value of the surface charge density. This is misleading, as all that can be said is that the field-density product has a certain value. So the labeling of both figures should be changed.

Small points and typos:

P.1 line 30: for completeness, dust also interacts via the semi-direct effect, whereby dust enhances cloud evaporation.

P. 8 line 257: presumably what is meant is that particles "would remain aloft", not "would be lofted"?

Fig. 5: the year cited should be 2003.

Fig. 7: In my view, the cumulative distribution shown is less clear than the alternative form of a differential distribution, and should be changed.

---

## Author Comment (AC1) · 30 Oct 2019

Thank you for your comments.

We will clarify the model description and what we mean by the field-density product.

We believe the value of 220 $\mu$C/m^2 is realistic for an upper bound of particle charge. We will clarify this in the text. Angus and Greber (2018) is a purely modeling study that found values on this same order of magnitude. Some of the papers they cite (such as Lowell and Rose Innes, 1980) show charge on this same order of magnitude.

For figures 6 and 7, the charge density follows a distribution. We can, instead of reporting the electric field as we did, report the standard deviation of the field-density product (charge of 110 $\mu$C/m^2). We do not believe the values to be unrealistic, as only

a small percentage (32%) of particles have charges greater than 110 $\mu$C/m^2.

We will make all the corrections to the paper, and ensure there are no more errors.

---

## Referee Comment (RC2) · Anonymous Referee #2 · 4 Dec 2019

I thank the authors for taking into account my previous comments and for their reply. Unfortunately, my main objection is still unanswered. The authors tried to apply the findings of their short time and small scale experiment, to the long transport of dust particles in the atmosphere. They assumed that the dust particle charge can obtain very large values, and the fair weather electric field, that can be sustained in several days, are able to create the necessary electrical force to counterbalance the gravitational force and transport the particles in large distances. I do not argue, that in the laboratory, very high values of surface charge density can appear, as a consequence of the turboelectric effect. My objection is that these high values are impossible to be met in the atmosphere. I will prove my point as follows.

The highest electric field value that can appear in the atmosphere has been measured in thunderstorms right before the initiation of a lightning flash. When the electric field exceeds a specific limit, the air is ionized that leads to the discharge of the charged particles and therefore to the reduction of the electric field. Up to now there has been no indication nor observation of lightning flashes in dust clouds. Therefore, it is a valid assumption that the electric field inside a dust cloud can never exceed the lightning initiation threshold. Let's assume this threshold to be equal to ±400 kV/m [*Stolzenburg et al.*, Geophys. Res. Lett., 34, 2007]. This threshold is scaled along the altitude as a function of the neutral density. Assuming the US Standard Atmosphere 1976, the vertical distribution of the electric field threshold is shown in Fig. 1.

[Figure]

Fig. 1: Vertical distribution of the $E_{thres}$.

The total electric field $E_{tot}$ at the surface of a spherical particle cannot exceed this threshold. This electric field can be written as $E_{tot} = E_{particle} + E_{ambient}$, where $E_{particle}$ is the electric field due to the

charge of the particle, and $E_{ambient}$ is the ambient electric field is the large scale electric field produced by other charged particles or other mechanisms (e.g. the potential difference between the Ionosphere and the Earth's suface).

The case of a positively charged particle inside a positive electric field (pointing upwards to the Ionosphere) is examined, but the methodology can be applied in any charge polarity and in any polarity of the electric field. Moreover, the particle is assumed as a conductor of electricity in electrostatics [*Ulanowski et al.*, Atmos. Chem. Phys., 7, 2007] .

Under these valid assumptions, the surface charge density $\sigma_{particle}$ of the particle, can never exceed the value $\sigma_{lim}=(E_{thres}-E_{ambient})\varepsilon_0$, where $\varepsilon_0$ is the vacuum permittivity. Fig. 2 and Fig.3 show the vertical distribution of the $\sigma_{lim}$ for positively and negatively charged particles, respectively. It is noted, that since an upward electric field is assumed, the electrical force for the positively charged particles tends to counterbalance the gravitational force, while the electrical force for the negatively charged particles acts in the same direction with the gravitational force leading to the faster fall of the particles (for an opposite polarity electric field obviously the opposite conclusion holds).

[Figure]

Fig.2: Vertical distribution of $\sigma_{lim}$ for positively charged dust particles.

[Figure]

Fig.3: Vertical distribution of $\sigma_{lim}$ for negatively charged dust particles.

In Fig. 2 and Fig. 3, the blue line corresponds to $E_{ambient} = 100$ V/m, the red line corresponds to $E_{ambient} = 1000$ V/m, the orange line corresponds to $E_{ambient} = 10$ kV/m, and the purple line corresponds to $E_{ambient} = 100$ kV/m.

If we are interested in the case that the electrical force counteracts the gravitational force, it is clear from Fig. 2 that the surface charge density is physically impossible to exceed the value of 3.5-3.6 $\mu C/m^2$. This is two orders of magnitude lower than the assumed upper limit value of the authors. Therefore, in their analysis in lines 230-240, the normal distribution of the surface charge density that has to be assumed for the long range transport, must have standard deviation s=3.5/3 $\mu C/m^2$, because 3s=3.5 $\mu C/m^2$. By doing that, the authors will realize that much stronger electric fields are required than the fair weather electric field, for the long range transport of the dust particles. Thus, the discussion section has to be modified accordingly.

---

## Author Response (AR1)

We would like to thank the reviewers for their helpful comments and suggestions. We have addressed each of their comments below and made appropriate changes in the paper.

Reviewer 1 comments:
COMMENT:
P.4 line 138 and following: the model description is cryptic and confusing, with the same symbols used for the mean values and standard deviations of three different quantities, diameter, charge density and initial vertical velocity. E.g. the variable described first as the standard deviation of the size distribution has the dimension of surface charge density (line 187). This makes it unnecessarily difficult to assess the methodology.

RESPONSE:
We have distinguished the different means and standard deviations that are associated with the three parameters. This change is seen in lines 137-139.

COMMENT:
Another potential confusion occurs in section 5: on p.7 lines 211-212 it is stated that the electrostatic forces were calculated as the product of the electric field and the surface charge density, meant here to be the same charge density as that on the particle surfaces. Taken at face value, this might be understood to imply that the atmospheric system is treated as a parallel capacitor, with the charge density equal to that found on particle surfaces (or on bulk quartz surfaces, as in one cited reference, Miura and Arakawa, 2007). This is misleading at best, and could potentially hide a basic error made by the authors - which is presumably not the case, judging from equation 2, which is correct.

Having said that, there is some justification for employing the field-density product, as both these parameters are unknown and the electrostatic force does depend on their product. So it should be explained clearly what is meant.

RESPONSE:
We have clarified the definition and usage of $\sigma E$ in lines 210 to 232.

COMMENT:
My main objection concerns the discussion of the results. It is concluded that even the fair weather electric could maintain large dust particles aloft. However, this is assuming extreme value of surface charge densities, 220 $\mu Cm^{-2}$, found in lab studies of bulk quartz surfaces (Miura and Arakawa, 2007), and untypically large (compare e.g. the values cited in Angus and Greber, 2018). More realistic values, especially for particulate matter rather than bulk material, should be cited from literature (which is not as sparse as the discussion implies, if spatial charge density measurements are included) and used in these conclusions. The authors own lab results point to charge densities two orders of magnitude lower than the above number, and similar values were found by Waitukaitis et al. (2014). So it should be concluded that electric fields exceeding the normal fair weather value are probably needed, as suggested previously by some authors. For example, it was estimated on the assumption of realistic spatial charge densities that fields of 2 kV/m would be required to maintain dust in suspension (Ulanowski et al., 2007). While such high values exceed the fair weather field, extensive literature suggests that they are not uncommon during dust episodes.

RESPONSE:

First, we have dialed-down our claims of atmospheric implications. For example, we replaced the sentence "Thus, our experimental and modelling results indicate that electrostatic forces should be considered when determining the effect of atmospheric dust on the climate" in the abstract with "Thus, our experimental and modelling results indicate that electrostatic forces may in some cases be relevant regarding the effect of atmospheric dust on the climate".

Second, we have rewritten the section from lines 210 to 232 and added extra detail to lines 282 to 304 to show that both the particle charge and electric field can vary substantially, and that the calculated value for $\sigma E$ of 38 mC V m$^{-3}$ is on the lower part of the given range of $\sigma E$. We now refer to ranges of values rather than specific values. We cite multiple well-regarded sources to justify the range of values.

COMMENT:
These unsupported conclusions are made prominent in Fig. 6 and 7, where the earlier convention to use the field-density product (as in Fig. 5) is unaccountably abandoned in favour of citing just the electric field. The hidden assumption, not mentioned in the figure captions, is the extreme, and probably unrealistic, value of the surface charge density. This is misleading, as all that can be said is that the field-density product has a certain value. So the labeling of both figures should be changed.

RESPONSE:

For figures 6 and 7, the charge density follows a distribution. We have edited the figures to show the standard deviation of $\sigma E$ used to perform each set of calculations with the charge density of 110 µC/m$^2$. As described above, we do not believe these values to be unrealistic. We have made edits in the text (lines 251 to 262 and 277) to go along with the changes to the figures.

COMMENT:
Small points and typos:
P.1 line 30: for completeness, dust also interacts via the semi-direct effect, whereby dust enhances cloud evaporation.
P. 8 line 257: presumably what is meant is that particles "would remain aloft", not "would be lofted"?
Fig. 5: the year cited should be 2003.
Fig. 7: In my view, the cumulative distribution shown is less clear than the alternative form of a differential distribution, and should be changed.

RESPONSE:

We have fixed these errors in the text, as well as checked thoroughly through to ensure that there are no more mistakes. We have also replaced figure 7 with a differential distribution rather than a cumulative distribution.

Reviewer 2 comments:

COMMENT:

I thank the authors for taking into account my previous comments and for their reply. Unfortunately, my main objection is still unanswered. The authors tried to apply the findings of their short time and small scale experiment, to the long transport of dust particles in the atmosphere. They assumed that the dust particle charge can obtain very large values, and the fair weather electric field, that can be sustained in several days, are able to create the necessary electrical force to counterbalance the gravitational force and transport the particles in large distances. I do not argue, that in the laboratory, very high values of surface charge density can appear, as a consequence of the turboelectric effect. My objection is that these high values are impossible to be met in the atmosphere. I will prove my point as follows.

The highest electric field value that can appear in the atmosphere has been measured in thunderstorms right before the initiation of a lightning flash. When the electric field exceeds a specific limit, the air is ionized that leads to the discharge of the charged particles and therefore to the reduction of the electric field. Up to now there has been no indication nor observation of lightning flashes in dust clouds. Therefore, it is a valid assumption that the electric field inside a dust cloud can never exceed the lightning initiation threshold. Let's assume this threshold to be equal to ±400 kV/m [*Stolzenburg et al.*, Geophys. Res. Lett., 34, 2007]. This threshold is scaled along the altitude as a function of the neutral density. Assuming the US Standard Atmosphere 1976, the vertical distribution of the electric field threshold is shown in Fig. 1.

[Figure]

Fig. 1: Vertical distribution of the $E_{thres}$.

The total electric field $E_{tot}$ at the surface of a spherical particle cannot exceed this threshold. This electric field can be written as $E_{tot} = E_{particle} + E_{ambient}$, where $E_{particle}$ is the electric field due to the charge of the particle, and $E_{ambient}$ is the ambient electric field is the large scale electric field produced by other charged particles or other mechanisms (e.g. the potential difference between the Ionosphere and the Earth's surface).

The case of a positively charged particle inside a positive electric field (pointing upwards to the

Ionosphere) is examined, but the methodology can be applied in any charge polarity and in any polarity of the electric field. Moreover, the particle is assumed as a conductor of electricity in electrostatics [*Ulanowski et al.*, Atmos. Chem. Phys., 7, 2007].

Under these valid assumptions, the surface charge density $\sigma_{particle}$ of the particle, can never exceed the value $\sigma_{lim}=(E_{thres}-E_{ambient})\varepsilon_0$, where $\varepsilon_0$ is the vacuum permittivity. Fig. 2 and Fig.3 show the vertical distribution of the $\sigma_{lim}$ for positively and negatively charged particles, respectively. It is noted, that since an upward electric field is assumed, the electrical force for the positively charged particles tends to counterbalance the gravitational force, while the electrical force for the negatively charged particles acts in the same direction with the gravitational force leading to the faster fall of the particles (for an opposite polarity electric field obviously the opposite conclusion holds).

[Figure]

Fig.2: Vertical distribution of $\sigma_{lim}$ for positively charged dust particles.

[Figure]

Fig.3: Vertical distribution of $\sigma_{lim}$ for negatively charged dust particles.

In Fig. 2 and Fig. 3, the blue line corresponds to $E_{ambient} = 100$ V/m, the red line corresponds to $E_{ambient} = 1000$ V/m, the orange line corresponds to $E_{ambient} = 10$ kV/m, and the purple line corresponds to $E_{ambient} = 100$ kV/m.

If we are interested in the case that the electrical force counteracts the gravitational force, it is clear from Fig. 2 that the surface charge density is physically impossible to exceed the value of 3.5-3.6 $\mu C/m^2$. This is two orders of magnitude lower than the assumed upper limit value of the authors. Therefore, in their analysis in lines 230-240, the normal distribution of the surface charge density that has to be assumed for the long range transport, must have standard deviation s=3.5/3 $\mu C/m^2$, because 3s=3.5 $\mu C/m^2$. By doing that, the authors will realize that much stronger electric fields are required than the fair weather electric field, for the long range transport of the dust particles. Thus, the discussion section has to be modified accordingly.

RESPONSE:

Thank you for the comments. We agree with the reviewer that the threshold charge density on a particle is limited by the threshold electric field from the particle that would cause a discharge into air. However, a complete analysis of this effect shows that the magnitude of particle charges we discuss in the paper in fact leads to electric fields below this threshold. Here, we expanded on the reviewer's back-of-the-envelope calculations to find the maximum surface charge density on particles based on Paschen's law, which describes the magnitude of the electric field threshold for gas breakdown.

The electric field from the particle that leads to gas discharge, $E_t$, is given by Paschen's law, which is a function of the distance from the particle surface and the ambient pressure,[4–7]

$$E_t = \frac{Bp}{\ln\left(\dfrac{A}{ln\left(\dfrac{1}{\gamma}+1\right)}\right) + \ln(pd)} \qquad (1)$$

where $p$ is the ambient pressure, $d$ is the distance from the surface, and $A$, $B$ and $\gamma$ are constants equal to 112.5 V (kPa cm)$^{-1}$, 2737.5 (kPa cm)$^{-1}$, and 0.01 respectively.[7]

The key point underlying Paschen's law is that gas breakdown occurs by an electron avalanche, wherein free electrons are accelerated by the electric field and undergo collisions with other molecules that release more free electrons, and each of these newly freed electrons are accelerated and collide to form even more free electrons, and so on.[8–11] In order for an electron avalanche to occur, the electric field must operate over a sufficiently long distance so as to enable a series of collisions to enable the avalanche to develop. For this reason, the threshold electric field diverges to infinity at small gap sizes.

We now address the maximum charge on for a spherical particle, as limited by Paschen's law. Importantly, we note that the electric field from the particle decreases as a function of distance from the particle surface, $d$, as given by Gauss's Law,

$$E = \frac{q}{4\pi\epsilon(R+d)^2} \qquad (2)$$

where $R$ is the radius of the particle, $q$ is the charge given by $q = 4\pi R^2 \sigma$ where $\sigma$ is the surface charge density of the particle. The maximum charge density possible before gas breakdown occurs when the electric field from the particle (Eq. 2) becomes larger than the electric field threshold from Paschen's law[8,11] (Eq. 1), noting that both of these quantities depend on the distance $d$ from the particle surface. As depicted in Figure 4, for a 10 µm particle, the maximum charge density prior to gas breakdown is 2800 µC/m$^2$; the maximum charge density on a particle is dependent on the particle size.

[Figure]

**Figure 4.** Electric field due to a 10 µm particle with different charge densities (dashed lines) and the threshold electric field given by Paschen's law (solid line)

Using this rigorous criterion, we obtain the maximum charge density for different particle sizes at a pressure of 54 kPa, characteristic of 5 km altitudes. As shown in Figure 5, see the maximum surface charge density is higher than 200 µC/m² at all relevant particle sizes.

[Figure]

**Figure 5.** Maximum surface charge density for different sized particles limited by gas breakdown described by Paschen's law (black) and Modified Paschen's law (red).

We note that at very small distances, recent work has shown that the breakdown electric field is more accurately represented by a Modified Paschen's law, which takes into account field emission.[5,6,11] As shown in Figure 5, the conclusions described above, regarding charge densities on the order of 200 $\mu$C/m$^2$, hold even when the Modifield Paschen's law is used in the analysis. The charge densities on surfaces of this order of magnitude and higher is supported by several experiments.[2,10,12–14] Thus, particle charge densities exceeding 200 $\mu$C/m$^2$ are physically possible.

We added details in lines 210 to 231 and lines 282 to 304.

(1)    Angus, J. C.; Greber, I.; Angus, J. C.; Greber, I. Tribo-Electric Charging of Dielectric Solids of Identical Composition Tribo-Electric Charging of Dielectric Solids of Identical Composition. **2018**, *174102*.

(2)    Lowell, J.; Rose-Innes, A. C. Contact Electrification. *Adv. Phys.* **1980**, *29* (6), 947–1023.

(3)    Waitukaitis, S. R.; Lee, V.; Pierson, J. M.; Forman, S. L.; Jaeger, H. M. Size-Dependent Same-Material Tribocharging in Insulating Grains. *Phys. Rev. Lett.* **2014**, *112*, 218001.

(4)    Fridman, A. *Plasma Chemistry*; Cambridge University Press: Cambridge, 2008.

(5)    Go, D. B.; Pohlman, D. A. A Mathematical Model of the Modified Paschen's Curve for Breakdown in Microscale Gaps. *J. Appl. Phys.* **2010**, *107* (10).

(6)    Go, D. B.; Venkattraman, A. Microscale Gas Breakdown: Ion-Enhanced Field Emission and the Modified Paschen's Curve. *J. Phys. D. Appl. Phys.* **2014**, *47* (50).

(7)    Raizer, Y. P. *Gas Discharge Physics*; Allen, J. E., Ed.; Springer-Verlag Berlin Heidelberg, 1991.

(8)    Matsuyama, T.; Yamamoto, H. Charge Relaxation Process Dominates Contact Charging of a Particle in Atmospheric Conditions. *J. Phys. D. Appl. Phys.* **1995**, *28* (12), 2418–2423.

(9)    Matsuyama, T.; Yamamoto, H. Charge-Relaxation Process Dominates Contact Charging of a Particle in Atmospheric Condition: II. the General Model. *J. Phys. D. Appl. Phys.* **1997**, *30* (15), 2170–2175.

(10)    Matsuyama, T.; Ogu, M.; Yamamoto, H.; Marijnissen, J. C. M.; Scarlett, B. Impact Charging Experiments with Single Particles of Hundred Micrometre Size. *Powder Technol.* **2003**, *135–136*, 14–22.

(11)    Matsuyama, T. A Discussion on Maximum Charge Held by a Single Particle Due to Gas Discharge Limitation. In *AIP Conference Proceedings*; American Institute of Physics Inc., 2018; Vol. 1927.

(12)    Horn, R. G.; Smith, D. T. Contact Electrification and Adhesion Between Dissimilar Materials. *Science (80-. ).* **1992**, *256* (5055), 362–364.

(13)    Hamamoto, N.; Nakajima, Y.; Sato, T. Experimental Discussion on Maximum Surface Charge Density of Fine Particles Sustainable in Normal Atmosphere. *J. Electrostat.* **1992**, *28* (2), 161–173.

[revised manuscript text omitted]